# Oversized cells activate global proteasome-mediated protein degradation to maintain cell size homeostasis

**Shixuan Liu[1,2,3]\*, Ceryl Tan[1,2], Chloe Melo-Gavin[1,2], Miriam B Ginzberg[2], Ron Blutrich[1,2], Nish Patel[2], Michael Rape[4], Kevin G Mark[4,5], Ran Kafri[1,2]\***

[1]Department of Molecular Genetics, University of Toronto, Toronto, Canada; [2]Cell Biology, The Hospital for Sick Children, Toronto, Toronto, Canada; [3]Department of Chemical and Systems Biology, Stanford University, Stanford, United States; [4]Department of Molecular Cell Biology, University of California at Berkeley, Berkeley, United States; [5]Department of Cell Biology, UT Southwestern Medical Center, Dallas, United States

**\*For correspondence:**
shixuan@stanford.edu (SL);
ran.kafri@sickkids.ca (RK)

**Abstract** Proliferating animal cells maintain a stable size distribution over generations despite fluctuations in cell growth and division size. Previously, we showed that cell size control involves both cell size checkpoints, which delay cell cycle progression in small cells, and size-dependent regulation of mass accumulation rates (Ginzberg et al., 2018). While we previously identified the p38 MAPK pathway as a key regulator of the mammalian cell size checkpoint (Liu et al., 2018), the mechanism of size-dependent growth rate regulation has remained elusive. Here, we quantified global rates of protein synthesis and degradation in cells of varying sizes, both under unperturbed conditions and in response to perturbations that trigger size-dependent compensatory growth slowdown. We found that protein synthesis rates scale proportionally with cell size across cell cycle stages and experimental conditions. In contrast, oversized cells that undergo compensatory growth slowdown exhibit a superlinear increase in proteasome-mediated protein degradation, with accelerated protein turnover per unit mass, suggesting activation of the proteasomal degradation pathway. Both nascent and long-lived proteins contribute to the elevated protein degradation during compensatory growth slowdown, with long-lived proteins playing a crucial role at the G1/S transition. Notably, large G1/S cells exhibit particularly high efficiency in protein degradation, surpassing that of similarly sized or larger cells in S and G2, coinciding with the timing of the most stringent size control in animal cells. These results collectively suggest that oversized cells reduce their growth efficiency by activating global proteasome-mediated protein degradation to promote cell size homeostasis.

## Editor's evaluation

This important study reports a previously undocumented role for ubiquitin-proteasome system (UPS)-mediated protein turnover in size control in human cells. The authors show that large cells undergo size compensation by actively reducing their rate of growth and this effect is shown to be mediated by an increase in the rate of proteasome-mediated degradation. The experiments are well controlled, and the conclusions of the study are compelling and well supported by the data. Overall the paper increases our knowledge related to size control mechanisms in dividing and non-dividing cells.

## Introduction

To maintain cell size homeostasis, proliferating cells double their mass from one division to the next and divide into similarly sized daughter cells. This stringent regulation of cell size exists in wide-ranging systems such as single-celled yeasts (*Hartwell et al., 1974*; *Johnston et al., 1977*; *Nurse, 1975*; *Schmoller et al., 2015*), animal cells in culture and within tissues (*Ginzberg et al., 2018*; *Liu et al., 2018*; *Xie and Skotheim, 2020*; *Zatulovskiy et al., 2020*), as well as plant meristem cells (*D'Ario et al., 2021*; *Serrano-Mislata et al., 2015*). Misregulation of cell size control results in increased size heterogeneity, which is typically observed during neoplastic growth and is a hallmark of malignancy in many tumors, including breast cancer and small cell lung cancer (*Asadullah et al., 2021*; *Bell and Waizbard, 1986*; *Ginzberg et al., 2015*; *Lee et al., 1992*).

How is cell size controlled and how does cellular growth in mass coordinate with the cell cycle program? A major mechanism is the cell size checkpoint, which hinders cell cycle progression for cells that are smaller than the target size. The mammalian G1/S cell size checkpoint was first reported by Zetterberg and Killander over 50 years ago (*Killander and Zetterberg, 1965*; *Zetterberg and Killander, 1965*). Subsequently, similar size checkpoints have been identified in single-celled yeasts (*Hartwell et al., 1974*; *Nurse, 1975*). In recent years, the functioning of cell size checkpoints has been investigated with increasingly powerful technologies (*Cadart et al., 2018*; *D'Ario et al., 2021*; *Liu et al., 2018*; *Perez-Gonzalez et al., 2019*; *Schmoller et al., 2015*; *Varsano et al., 2017*; *Xie and Skotheim, 2020*). With increased resolution, these newer studies directly confirmed that smaller cells spend longer periods of growth in G1, allowing cells to reach the target size as they transition into S phase. Although it is not yet clear how animal cells sense their size, studies have revealed roles for both the p38 MAPK (*Liu et al., 2018*; *Sellam et al., 2019*) and the CDK4/Rb pathways (*Datar et al., 2000*; *Tan et al., 2021*; *Zatulovskiy et al., 2020*).

In addition to cell size checkpoints, two independent studies suggest that mammalian cells also employ size-dependent regulation of cellular growth rate (*Cadart et al., 2018*; *Ginzberg et al., 2018*). Authors of the studies used different methods to measure the growth of individual cell size throughout the cell cycle and found that the rate of cell growth negatively correlated with cell size at various cell cycle stages. *Ginzberg et al., 2018* further applied chemical and genetic perturbations that slowed down or accelerated cell cycle progression and observed compensatory changes in the rate of cellular growth, which buffered the initial effect on cell size (Figure 1A). For example, cells under CDK2 inhibition were forced to grow for a longer period in G1, resulting in an immediate increase in cell size. Remarkably, cells later compensated for this longer period of growth with slower rates of mass accumulation, resulting in only a small increase in cell size (Figure 1—figure supplement 1). To discriminate this mechanism from *size checkpoints*, we use the term *size-dependent compensatory growth*. Notably, this compensation does not occur with CDK4/6 inhibition (*Ginzberg et al., 2018*), which was found to reprogram cells to a larger target size (see *Tan et al., 2021*). Altogether, these studies suggest that cells sense their size to regulate not only cell cycle progression (e.g., cell cycle checkpoints), but also adapt the cell growth program to maintain cell size homeostasis.

## Results

In this study, we ask whether the size-dependent regulation of cellular growth is mediated by protein synthesis or protein degradation. We found that proteasome-mediated global protein degradation, rather than protein synthesis, underlies the size-dependent compensatory growth and functions in parallel with cell size checkpoints to promote cell size homeostasis.

To reliably induce a compensatory growth slowdown, we utilized a CDK2 inhibitor assay previously established by *Ginzberg et al., 2018*. We applied this assay to human retinal pigment epithelial-1 (RPE1) cells, a well-characterized non-transformed epithelial cell line. In this assay, unsynchronized cell populations were treated with a low dose of the CDK2 inhibitor, SNS-032, to induce a longer G1 phase. We optimized a concentration range that inhibits CDK2 function without arresting the cell cycle (*Figure 1—figure supplement 1*). Cells were subsequently fixed at different timepoints and profiled for proliferation rate, cell size, and cell cycle stage. Cell size was measured with Alexa fluorophore-conjugated succinimidyl ester (SE), which reacts with primary amines and quantifies total protein content of the cell as previously described (*Kafri et al., 2013*; *Mugahid et al., 2020*; *Neurohr et al., 2019*). As shown by *Ginzberg et al., 2018*, the temporal influence of CDK2 inhibition on cell

size is characterized by a two-stage process: an early response followed by a delayed compensation. Although cell size initially increased for CDK2-inhibited cells as a result of the prolonged cell cycle, it gradually plateaus after ~24 hr due to a compensatory slowdown in growth (*Figure 1—figure supplement 1*). Average rates of cell proliferation and cell growth can be inferred from the dynamics of cell number and cell mass (see 'Materials and methods'). This revealed that CDK2-inhibited cells initially grew at the same rate as control but later compensated with a 24% reduced rate of mass accumulation (*Figure 1—figure supplement 1*). In comparison, rates of cell proliferation remained unchanged (25% lower than control) throughout the experiment (*Figure 1—figure supplement 1*; *Ginzberg et al., 2018*). The delayed compensatory slowdown in cellular growth suggests that the influence of CDK2 inhibitor on growth rate is indirect and is mediated by a property that accumulates over time, presumably cell size.

What underlies this size-dependent regulation of cell growth? The rate of macromolecular growth depends on the interplay between the biosynthesis and degradation of proteins, lipids, polysaccharides, and other macromolecules (*Alber and Suter, 2019*). In an actively proliferating mammalian cell, proteins represent more than half of the cell's total dry mass (*Mitchison, 1971*) and are under active turnover (*Ghenim et al., 2021*; *Liu et al., 2020*). Total protein content also linearly scales with the dry mass and volume of cells at different cell cycle stages (*Berenson et al., 2019*; *Kafri et al., 2013*). The balance between protein translation and degradation has been shown to significantly influence cell size in various cell types, including neurons, muscles, and cancer cells (*Acebron et al., 2014*; *Franklin and Johnson, 1998*; *Gordon et al., 2013*; *Sandri, 2013*), suggesting a vital role of protein homeostasis in cell size control.

## A quantitative assay for the size-dependent regulation of cellular growth rate

To further investigate the robustness of the assay, we employed time-lapse imaging to directly quantify single-cell dynamics of size growth. Using the nuclear area delineated by SiR-DNA as a proxy for cell size, we followed RPE1 cells stably expressing the degron of Geminin fused to a monomeric Azami green (mAG-hGem) (*Sakaue-Sawano et al., 2008*) throughout ~60 hr of SNS-032 treatment. Consistent with results from the fixed cell assay, CDK2 inhibition resulted in a longer G1 phase (+3.5 hr, 32%) and lower growth rates of nuclear size (–21%) (*Figure 1B–E*). Although CDK2-inhibited cells had a larger initial size as a result of prolonged growth duration from the previous cycle, they grew slower and became similarly sized as control cells around the G1/S transition, which was apparent when growth trajectories were computationally synchronized to the timing of their G1/S transition (*Figure 1E*). These findings provide further evidence for the size-dependent compensation in cell growth. CDK2 inhibition induced prolonged growth duration and an initial increase in cell size, which is then compensated by a delayed response of reduced growth rate, affirming that cellular growth rate is adaptively regulated to maintain cell size homeostasis.

## Compensatory changes in cellular growth rate are not regulated at the level of global protein synthesis

Because cell growth reflects the balance between rates of protein synthesis and protein degradation, we first asked whether the compensatory growth slowdown is mediated through reduced rates of protein synthesis in large cells. We used a multiplex, single-cell labeling strategy to jointly profile overall protein synthesis rates, macromolecular protein mass (by SE), and cell cycle state in thousands of asynchronized proliferating cells. To quantify global translation rates, we performed kinetic pulse measurements with a derivatized methionine analog, L-azidohomoalanine (AHA) (*Calve et al., 2016*), which measures the amount of AHA that is incorporated into newly synthesized proteins. We performed an AHA pulse for 3 hr to obtain a high signal-to-noise ratio in a relatively short period of the cell cycle. Treating cells with the protein synthesis inhibitor, cycloheximide (CHX), significantly reduces AHA incorporation (*Figure 2A*), confirming the efficacy of the assay.

If the compensatory slowdown of growth is driven by slower rates of protein synthesis, larger cells should have lower rates of AHA incorporation. Our results, however, demonstrate the contrary. In unperturbed cells, AHA incorporation levels positively correlated with total protein content ($R = 0.91$) (*Figure 2A*), indicating a faster rate of protein synthesis in large cells. To test if the positive correlation between cell size and AHA incorporation is cell cycle dependent, we further segregated cells by

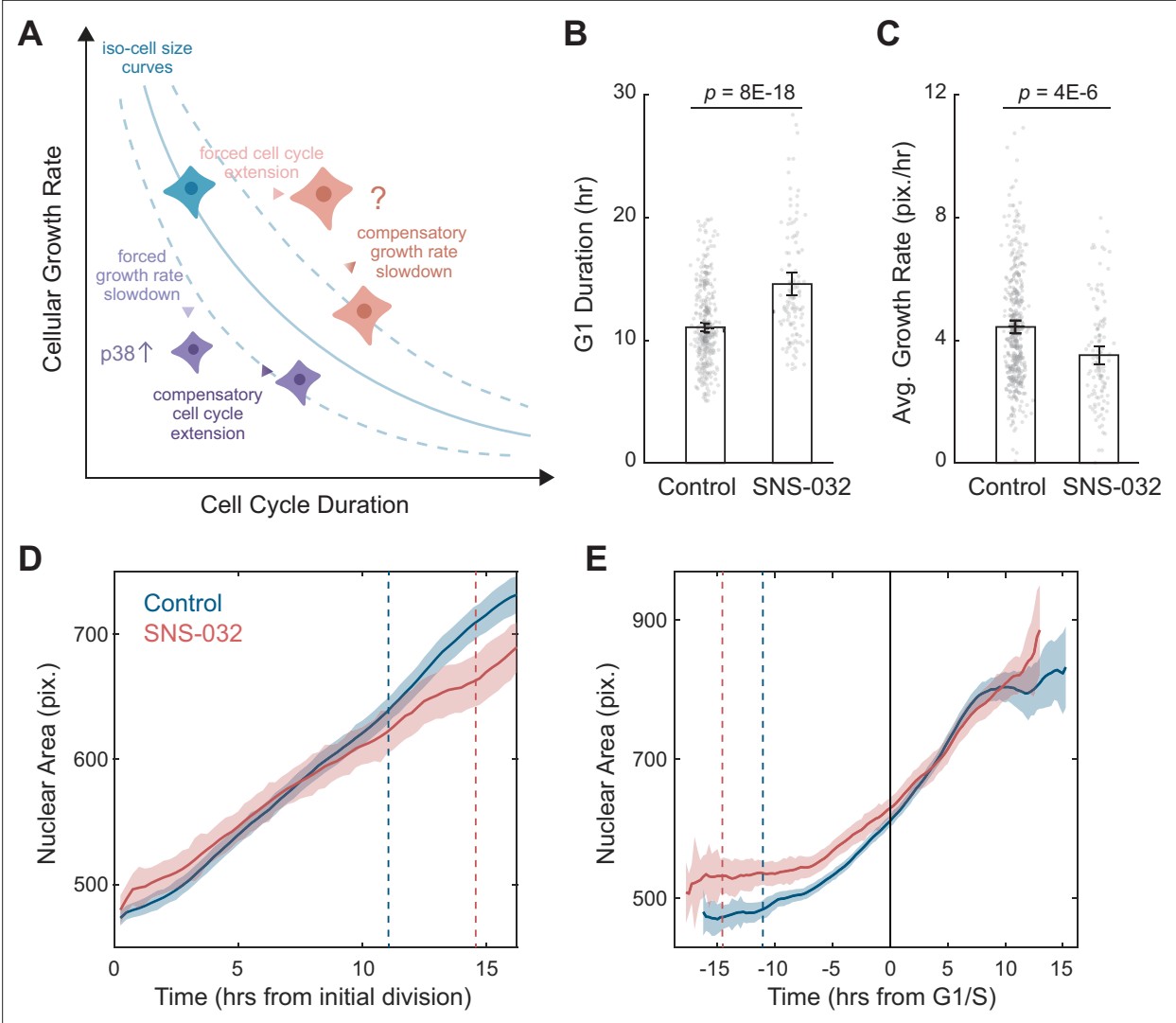

**Figure 1.** A quantitative assay for the size-dependent regulation of cellular growth rate. (**A**) Schematic showing the model of cell size control as described by *Ginzberg et al., 2018*. The solid blue line represents an iso-cell size curve: combinations of cellular growth rates (mass accumulation rates) and cell cycle length that result in the same cell size. The dashed blue lines represent iso-size curves at larger (upper curve) or smaller (lower curve) cell sizes. Perturbations that decrease cellular growth rate (e.g., mTOR inhibition) result in an initial decrease in cell size, followed by an adaptation involving the activation of p38 MAPK pathway (*Liu et al., 2018*) to prolong the cell cycle, preventing further decrease in size (purple cells). Perturbations that lengthen the cell cycle (e.g., CDK2 inhibition) result in an initial increase in cell size, followed by a compensatory slowdown of growth rate that prevents further increase in cell size (red cells). (**B, C**) Bar plots comparing the G1 duration (**B**) and average growth rate (**C**) for control (0.1% v/v DMSO, N = 365 single-cell tracks) and CDK2-inhibited RPE1 cells (20 nM SNS-032, N = 125), measured by time-lapse live-cell imaging. Nuclear area is measured as a proxy of cell size. Error bars represent mean ± 95% CIs. p-values are calculated using two-sample *t*-test. (**D, E**) Average nuclear area as a function of time for control and CDK2-inhibited cells, with growth trajectories either synchronized to the first division/birth (**D**) or the time of G1/S transition (**E**). Shaded areas mark 95% CIs. In (**D**), dashed lines mark the average time of G1/S transition. In (**E**), the solid black line marks the time of G1/S transition, and dashed lines mark the average time of first birth/division. Note that CDK2-inhibited cells had larger initial sizes, progressed through G1/S later, and showed slower growth in size (i.e., shallower slope), compared to the control. Also see *Figure 1—figure supplement 1*.

The online version of this article includes the following source data and figure supplement(s) for figure 1:

**Source data 1.** File contains the source code and source data necessary to generate *Figure 1* using Matlab.

**Figure supplement 1.** RPE1 cells under CDK2 inhibition compensate for prolonged growth with slower rates of mass accumulation.

**Figure supplement 1—source data 1.** File contains the source code and source data necessary to generate *Figure 1—figure supplement 1* using Matlab.

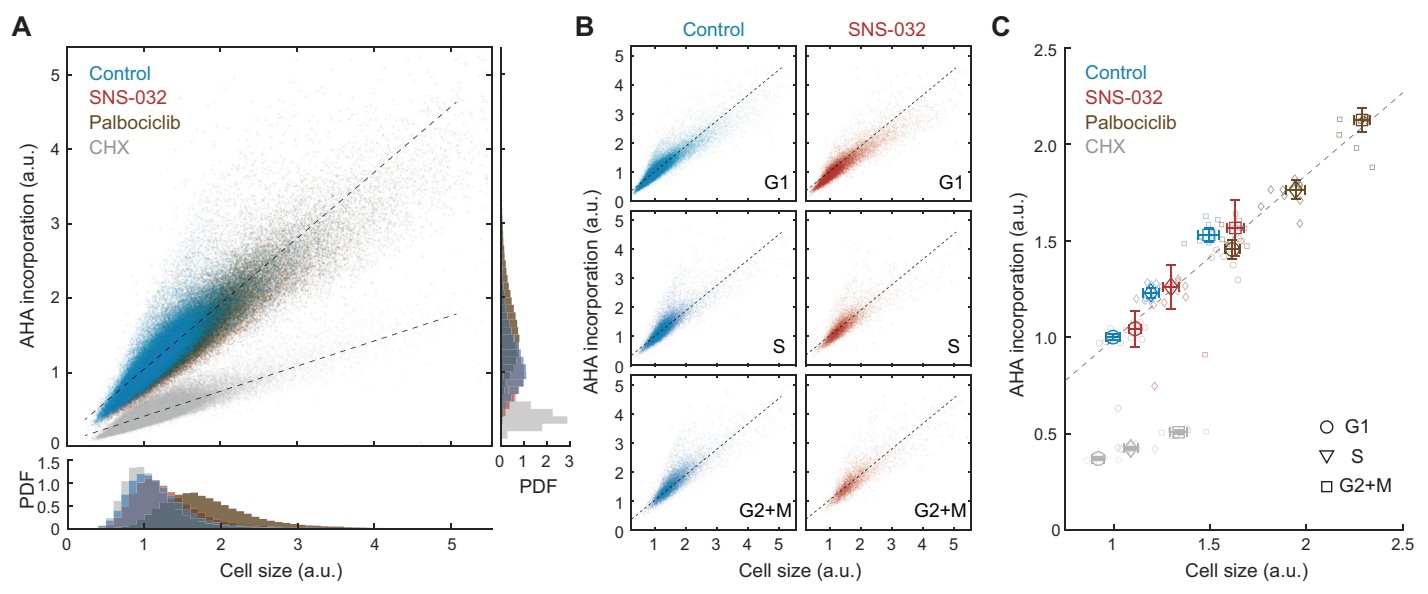

**Figure 2.** Global rates of protein synthesis scale with cell size. (**A**) Scatterplot showing single-cell measurements of global protein synthesis rates (AHA incorporation) as a function of cell size (SE) for control (0.1% v/v DMSO) and cells under 48 hr treatment of the protein synthesis inhibitor cycloheximide (CHX, 3 μM), the CDK2 inhibitor SNS-032 (25 nM), and the CDK4/6 inhibitor palbociclib (50 nM). Lines show linear fit. Bottom and right panels show histograms of cell size and AHA incorporation, respectively. (**B**) Scatterplots as in panel (**A**) plotted separately for G1, S, and G2+M phase cells under control or CDK2 inhibition. (**C**) Median rates of protein synthesis for G1, S, and G2+M cells under each of the experimental conditions. Data presented as fold-change over the average of control cells in G1, with error bars indicating ±95% CI, N = 10 replicate wells.

The online version of this article includes the following source data for figure 2:

**Source data 1.** File contains the source code and source data necessary to generate *Figure 2* using Matlab.

their cell cycle stage and found that the dependence of AHA incorporation on size was consistent for G1, S, and G2 cells (*Figure 2B*). In all cell cycle stages, larger cells incorporated more AHA per unit time. Markedly, the linear dependence of AHA incorporation on cell size persists across a wide range of sizes, even for cells that are approximately twofold larger, as induced by CDK4/6 inhibition (*Figure 2C*).

Because we did not observe a compensatory slowdown of AHA incorporation in naturally large cells, we next examined whether the compensatory growth slowdown following CDK2 inhibition is regulated at the level of protein synthesis. We employed the same strategy described in *Figure 1—figure supplement 1* and additionally measured rates of AHA incorporation. CDK2-inhibited cells demonstrated a slight increase in both cell size (~11%) and AHA incorporation (~4%) compared to that of control (*Figure 2C*), despite the compensatory slowdown of growth. Consistent with measurements in unperturbed cells, CDK2-inhibited cells maintained a similar linear correlation between cell size and AHA incorporation, both at the single-cell level and across different cell cycle stages (*Figure 2B and C*), suggesting that the protein synthesis machinery is not affected by CDK2 inhibition throughout the duration of the experiment. Together, results from the AHA pulse experiments suggest that the compensatory growth slowdown in CDK2-inhibited cells is not mediated through a decrease in overall rates of protein synthesis but likely through an increase in overall protein degradation.

## Large cells have higher rates of global protein degradation

To measure rates of protein degradation, we implemented a CHX chase assay and quantified the loss of total protein mass over time. In this assay, protein synthesis was inhibited by CHX; therefore, the loss of macromolecular protein mass reflected changes resulting from protein degradation. Interestingly, we observed significantly increased rates of protein degradation in larger cells (80th percentile) compared to smaller cells (20th percentile) (*Figure 3A*, *Figure 3—figure supplement 1*), supporting the hypothesis that large cells may activate global protein degradation to initiate a compensatory growth slowdown.

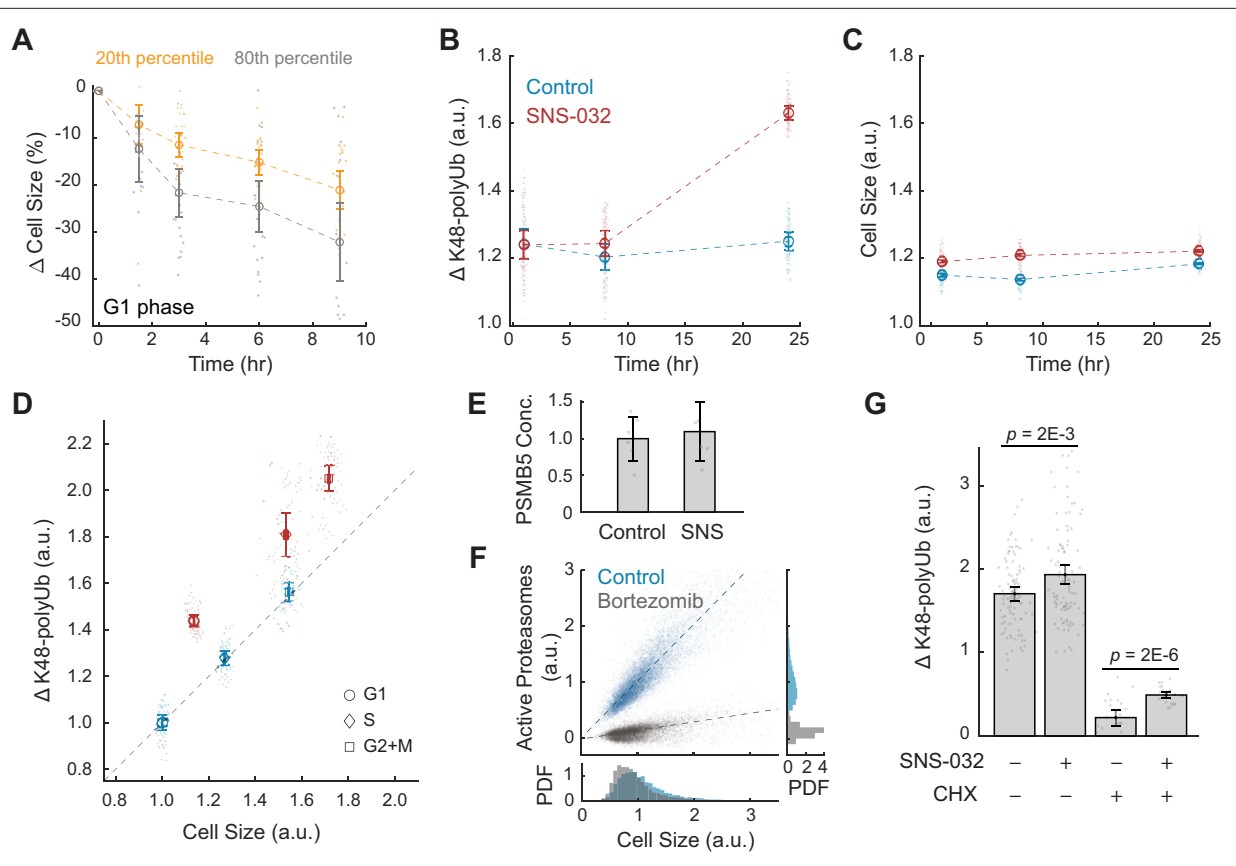

**Figure 3.** Large cells have higher rates and efficiency of global protein degradation. (**A**) Change in cell size measured by SE for small (20th percentile) and large (80th percentile) G1 cells following treatment with the protein synthesis inhibitor cycloheximide (CHX, 10 µM) compared to time point 0 (0.1% v/v DMSO). Cell size changes are expressed as the fraction of the median size at time point 0. Error bars represent mean ± 95% CI (N = 22 replicate wells). (**B, C**) Proteasome-mediated global protein degradation rate (ΔK48-polyUb, **B**) and cell size (SE, **C**) over time in control (0.1% v/v DMSO) and in cells treated with the CDK2 inhibitor SNS-032 (25 nM) for 1, 8, and 24 hr. To measure ΔK48-polyUb, cells were immunostained with a K48 linkage-specific polyubiquitin antibody that quantifies the total K48-linked polyubiquitinated proteins (K48-polyUB). ΔK48-polyUb is defined as the clearance rate of K48-polyUb, calculated as the difference between K48-polyUB measured with or without a short-term (30 min) treatment of the proteasome inhibitor Carfilzomib (CFZ, 8 µM) before fixation. Data presented as mean ± 95% CI, N = 9 replicate wells. (**D**) ΔK48-polyUb as a function of cell size (SE) for control and CDK2-inhibited cells (25 nM SNS-032, 24 hr), separated by cell cycle stage. Values are normalized to the average of control cells in G1. Data presented as mean ± 95% CI, N = 9 replicate wells. Right-tailed two-sample *t*-tests indicate significant increase in ΔK48-polyUb of the SNS-032-treated cells compared to the control across the cell cycle: p-values = 7E-95 (G1), 3E-62 (S), and 3E-72 (G2+M). Dashed line represents 1:1 proportional changes in ΔK48-polyUb and cell size (y = x). Note the proportional changes in ΔK48-polyUb and size for control and disproportionately higher increases in ΔK48-polyUb compared to size for SNS-032 treated cells. (**E**) Quantification of PSMB5 concentrations from six replicates of western blots (see ***Figure 3— figure supplement 5***), presented as fold-change of control. Error bars present mean ± 95% CI. Western blots were loaded with the same amount of cell lysates, thus data shown here reflect PSMB5 levels normalized by cell size. (**F**) Scatterplot of single-cell measurements of total proteasome activity (MV151) as a function of cell size (SE) for control (0.1% v/v DMSO) and bortezomib-treated (1 µM) cells, along with histograms showing distribution of cell size (bottom) and proteasome activity (right). Both measurements are shown as relative values to the average of the control. Note the level of active proteasomes may be negative as the quantification was performed after subtraction of background. Lines show linear fit. (**G**) ΔK48-polyUb for control (0.1% v/v DMSO) and SNS-032 (25 nM, 24 hr) treated cells with or without a short-period (3 hr) treatment of the protein synthesis inhibition CHX (3 µM) before fixation, shown for cells at the G1/S transition. Data presented as mean ± 95% CI. N = 5 and 10 replicate wells, respectively, for conditions with and without CHX. See ***Figure 3—figure supplement 6*** for cells at other cell cycle stages.

The online version of this article includes the following source data and figure supplement(s) for figure 3:

**Source data 1.** File contains the source code and source data necessary to generate ***Figure 3A***, ***Figure 3—figure supplement 1*** using MATLAB.

**Source data 2.** File contains the source code and source data necessary to generate ***Figure 3B–D***, ***Figure 3—figure supplement 2***; ***Figure 3—figure supplements 3 and 4*** using MATLAB.

**Source data 3.** File contains the source code and source data necessary to generate ***Figure 3E–F***, ***Figure 3—figure supplement 5B*** using MATLAB.

**Source data 4.** File contains the source code and source data necessary to generate ***Figure 3G***, ***Figure 3—figure supplement 6*** using MATLAB.

**Figure supplement 1.** Large cells show increased protein degradation throughout the cell cycle.

*Figure 3 continued*

**Figure supplement 2.** Short treatment of the proteasome inhibitor carfilzomib (CFZ) does not affect cell size or fraction of cells in different cell cycle stages.

**Figure supplement 3.** CDK2 inhibition increases K48-polyUb turnover across cell cycle stages.

**Figure supplement 4.** Changes in cell size and protein degradation rates in response to other cell cycle perturbations.

**Figure supplement 5.** CDK2 inhibition does not significantly affect proteasome content per unit mass.

**Figure supplement 5—source data 1.** File contains uncropped and labeled gels for *Figure 3—figure supplement 5A*.

**Figure supplement 5—source data 2.** File contains raw unedited gels for *Figure 3—figure supplement 5A*.

**Figure supplement 6.** Size-dependent compensatory degradation involves nascently translated and long-lived proteins.

Proteasome-mediated protein degradation is a major route of protein turnover in non-starved cells (*Lecker et al., 2006*). The major targeting signal for proteasomal degradation is a post-translational modification involving ubiquitin chains linked through Lysine 48 (K48) (*Yau and Rape, 2016*). To further test whether larger cells undergo higher rates of proteasome-mediated degradation, we measured the turnover (i.e., clearance) of K48-linked polyubiquitinated proteins (K48-polyUb) in CDK2 inhibited cells that undergo a compensatory growth slowdown. K48-polyUb turnover rates were quantified as the increase in intracellular pools of K48-polyUb that is caused by a short (30 min) treatment with a potent irreversible proteasome inhibitor, carfilzomib (CFZ). The excess amount of K48-polyUb in CFZ-treated cells compared to that of untreated cells, which we term ΔK48-polyUb, estimates the amount of K48-polyUb proteins that would have been degraded during the 30 min had the proteasome not been inhibited. Cell size and the partitioning of cells into different cell cycle stages were not significantly influenced by the 30 min treatment with CFZ (*Figure 3—figure supplement 2*). Using this method, we measured the rates of proteasome-mediated protein degradation (ΔK48-polyUb) in CDK2-inhibited cells that were induced to undergo a compensatory slowdown of growth. CDK2-inhibited cells showed higher rates of ΔK48-polyUb, affirming a role of proteasomes in compensatory growth. The dependence of compensatory growth on proteasomal degradation is further supported by the temporal similarity of these two processes. Both the slowdown in growth rate and increase in ΔK48-polyUb showed delayed dynamics following CDK2 inhibition. CDK2-inhibited cells had similar ΔK48-polyUb as control during the first 8 hr of treatment, but later showed a significant increase in ΔK48-polyUb at 24 hr, indicating increased rates of protein degradation (*Figure 3B and C*). The enhanced turnover of K48-polyUb proteins aligns temporally with the compensatory growth rate changes observed in CDK2-inhibited cells (*Figure 1—figure supplement 1*). This pattern was consistently observed across G1, S, and G2+M phases in CDK2-inhibited cells (*Figure 3—figure supplement 3*). The delayed kinetics suggest that the increased ΔK48-polyUb is not triggered by the immediate inhibition of CDK2 activity but rather from the buildup of excessive cell mass triggered by the gradual influence of the longer cell cycle. Altogether, these results indicate that the compensatory slowdown of cell growth is mediated by increased clearance of K48-polyUb.

## Large cells under compensatory growth slowdown have higher rates of protein degradation per unit cellular mass

Our findings suggest that larger cells have higher rates of protein synthesis as well as higher rates of protein degradation. A simple interpretation of these data would posit that both the rates of protein synthesis and degradation linearly scale with cell size. For example, a cell that increased 50% in size would have a 50% increase in both the rates of protein synthesis and degradation. To examine this quantitative relationship, we calculated fold changes in protein synthesis and degradation as a function of fold changes in cell size. Perturbation of CDK2 activity promotes proportional increases in both cell size and the rate of AHA incorporation across all cell cycle stages (*Figure 2C*). However, CDK2-inhibited cells exhibit disproportionately larger changes in protein degradation (ΔK48-polyUb) relative to the changes in cell size, suggesting a superlinear relationship. On average, ΔK48-polyUb increased by ~30% in CDK2-inhibited cells, whereas mean cell size only increased by ~3.4% compared to control (*Figure 3B and C*). This disproportionate increase in degradation rates was consistent across all interphase stages (G1: 44%; S: 42%; G2+M: 31%) (*Figure 3D*). These observations highlight an interesting contrast between the cell size dependencies of protein synthesis and degradation. Although larger cells have higher absolute rates of protein synthesis, the rates of AHA incorporation

per unit cell mass remain relatively constant across different cell sizes. In contrast, larger cells have higher rates of protein degradation even when quantified per unit cellular mass, suggesting an activation of the protein degradation pathways in large cells, in addition to large cells having more proteins to degrade.

To explore the specificity of these results, we then examined protein degradation in cells treated with the CDK4/6 inhibitor palbociclib, which induces a larger target size without compensatory growth slowdown (*Ginzberg et al., 2018*; *Tan et al., 2021*). As expected, CDK4/6 inhibition resulted in significant increases in both cell size (G1: 32%; S: 69%; G2+M: 38%) and ΔK48-polyUb (G1: 30%; S: 66%; G2+M: 45%) at 24 hr of treatment (*Figure 3—figure supplement 4*). However, the increases in ΔK48-polyUb remained proportional to the changes in cell size. Unlike the superlinear increase in ΔK48-polyUb observed in CDK2-inhibited cells (*Figure 3D*), cell size and ΔK48-polyUb under CDK4/6 inhibition followed the same linear relationship as control cells across different cell cycle stages (*Figure 3—figure supplement 4*). This suggests that compensatory degradation is not triggered merely by shifts in target size.

Next, we examined additional cell size perturbations to test whether the upregulated protein degradation during compensatory growth slowdown is a general size control mechanism or respond specifically to CDK2 inhibition and prolonged G1. Using data from a previous in-house chemical screen (*Kaufman et al., 2022*), we identified compounds that increase cell size by altering progression rates of different cell cycle stages: JNJ-26854165, which slows S phase progression, and barasertib, which delays mitosis (*Figure 3—figure supplement 4*). Importantly, the two compounds do not directly disrupt DNA or cytoskeleton integrity (*Bavetsias and Linardopoulos, 2015*; *Jones et al., 2013*). Both compounds led to increased global protein degradation, as measured by ΔK48-polyUb (*Figure 3—figure supplement 4*). Similar to the effects of CDK2 inhibition, they induced disproportionately higher increases in ΔK48-polyUb relative to the cell size increases, particularly in the affected cell cycle stages. Barasertib-treated G2+M cells exhibited a 10% increase in cell size but a 37% increase in ΔK48-polyUb at 8 hr of treatment, and the superlinear increase in ΔK48-polyUb persisted at 24 hr of treatment (*Figure 3—figure supplement 4*). JNJ-26854165-treated S phase cells showed a temporary superlinear increase in degradation at 8 hr but not at 24 hr of treatment (*Figure 3—figure supplement 4*). Taken together, these results suggest that activation of protein degradation serves as a general compensatory mechanism of cell size control in response to prolonged growth periods. Whereas the dynamics of the compensatory degradation may vary, it can be triggered irrespective of the specific cell cycle stage affected.

## Compensatory growth slowdown involves increased proteasome activity in degrading both newly synthesized and long-lived proteins

What mediates the size-dependent increase in proteasome-mediated protein degradation? One possibility is that large cells have more proteasomes. To test this, we compared proteasome concentration in control and CDK2-inhibited cells by probing the 20S proteasome subunit β5, PSMB5 (*Russell et al., 1999*). These measurements showed that CDK2-inhibited cells had a similar level of proteasomes per unit mass as control (*Figure 3E*, *Figure 3—figure supplement 5*), suggesting that the total number of proteasomes per cell increases proportionally with cell size in CDK2-inhibited cells. Next, we tested if large cells may activate more proteasomes. It was reported that a fraction of the intracellular proteasomes are catalytically inactive and can be stimulated in response to stress (*Collins and Goldberg, 2017*; *Livneh et al., 2016*). To quantify the levels of active proteasomes in single cells, we used a proteasome activity probe, MV151. MV151 is a fluorescent and cell-permeable proteasome inhibitor that selectively binds to the catalytic core of active proteasomes and provides fluorescence readouts of active proteasomes at single-cell resolution (*Verdoes et al., 2006*). These data revealed a linear correlation between cell size and levels of active proteasomes ($R = 0.90$), suggesting that large cells have proportionately higher levels of active proteasomes (*Figure 3F*). As a positive control, the proteasome inhibitor, bortezomib, eliminated the dependence of active proteasomes on cell size. The observation that larger cells have proportionally higher levels of active proteasomes, yet exhibit a superlinear increase in K48-polyUb degradation, suggests that the efficiency of protein degradation per active proteasome may be enhanced, warranting further investigation. An increased degradation efficiency might be achieved through increased efficiency in protein ubiquitination, such as activating E3 ubiquitin ligases that target proteins for degradation.

What proteins are degraded during compensatory growth slowdown? Previous studies have shown that nascent proteins, or newly synthesized polypeptides, are actively degraded in eukaryotic cells as part of protein quality control (*Wolff et al., 2014*). It has been estimated that up to 15% of nascent chains in human cells are tagged for degradation (*Wang et al., 2013*). To examine if nascent or long-lived proteins are preferentially degraded in large cells, we used CHX to acutely inhibit protein synthesis and then measured K48-polyUb turnover. The comparison between cells with and without CHX treatment allowed distinction between the degradation of nascent (CHX-sensitive) and non-nascent (CHX-insensitive) proteins. As anticipated, CHX treatment significantly reduced ΔK48-polyUb levels (*Figure 3G*, *Figure 3—figure supplement 6*), confirming that nascent proteins contribute substantially to proteasomal degradation (*Wang et al., 2013*). We then compared cells with or without CDK2 inhibition, the difference of which reflects the level of size-related compensatory degradation. In the absence of CHX, CDK2 inhibition resulted in significant increases in ΔK48-polyUb across the cell cycle (*Figure 3G*, *Figure 3—figure supplement 6*), consistent with previous results. However, in the presence of CHX, the increase in ΔK48-polyUb was primarily observed at the G1/S transition (*Figure 3G*, *Figure 3—figure supplement 6*). These results suggest that both nascent and long-lived proteins are targeted for degradation during compensatory growth slowdown, with long-lived proteins playing a crucial role at the G1/S transition.

## Large cells at the G1/S transition show hyperactive global protein degradation

To further dissect the cell cycle dependency of the increased global protein degradation observed in large cells, we compared ΔK48-polyUb levels across different cell cycle stages. Strikingly, cells at the G1/S transition exhibit the highest ΔK48-polyUb levels in both control and CDK2-inhibited cells (*Figure 4A and B*). Remarkably, rates of global protein degradation observed in these large G1/S cells surpassed those of similarly sized or even larger cells in S and G2 phases, suggesting a hyperactivation of global protein degradation at the G1/S transition.

To better characterize the cell cycle dynamics of compensatory degradation, we performed pseudotime trajectory analysis and aligned cells to a continuous cell cycle axis based on their DNA content and the FUCCI cell cycle marker mAG-hGem (*Figure 4C*, see 'Materials and methods'; *Kafri et al., 2013*). This method allowed extraction of average cellular dynamics from single-cell snapshots of fixed steady-state populations (*Kafri et al., 2013*). Interestingly, the analysis revealed a subpopulation of exceptionally large cells at the G1/S transition, often comparable in size or even larger than cells in S and G2 phases (*Figure 4D*). Time-lapse imaging confirmed that large G1/S cells were not arrested but continued to progress through the cell cycle (*Figure 4—figure supplement 1*). These large G1/S cells exhibited high levels of ΔK48-polyUb per cell (*Figure 4—figure supplement 1*), and importantly, the highest levels of ΔK48-polyUb per unit mass (*Figure 4E*), further supporting the notion of hyperactive protein degradation in large G1/S cells.

To further test this observation, we used time-lapse imaging and examined the impact of partial protein synthesis inhibition on cellular growth rates before and after the G1/S transition. We applied mTOR inhibition and measured cellular growth rates separately for small and large cells. Under control conditions, large cells on average grew faster than small cells, both before and after the G1/S transition (*Figure 4—figure supplement 2*), and when averaged across the entire cell cycle (*Figure 4—figure supplement 2*), likely due to their higher protein synthesis rates (*Figure 2*, *Figure 4—figure supplement 2*). However, upon mTOR inhibition, the cellular growth rates of small and large cells converged before the G1/S transition but not after (*Figure 4—figure supplement 2*). This convergence occurred despite a proportional decrease in protein synthesis rates by mTOR inhibition for all cells, with large cells continuing to synthesize proteins faster than small cells (*Figure 4—figure supplement 2*). These results provide additional evidence that large cells prior to the G1/S transition exhibit elevated rates of protein degradation, which counterbalance the higher protein synthesis rates in large cells.

In summary, our results demonstrate that large cells exhibit elevated rates of protein degradation throughout interphase, with a particularly pronounced increase in large cells at the G1/S transition. This mechanism may function in conjunction with the cell size checkpoints to promote cell size uniformity.

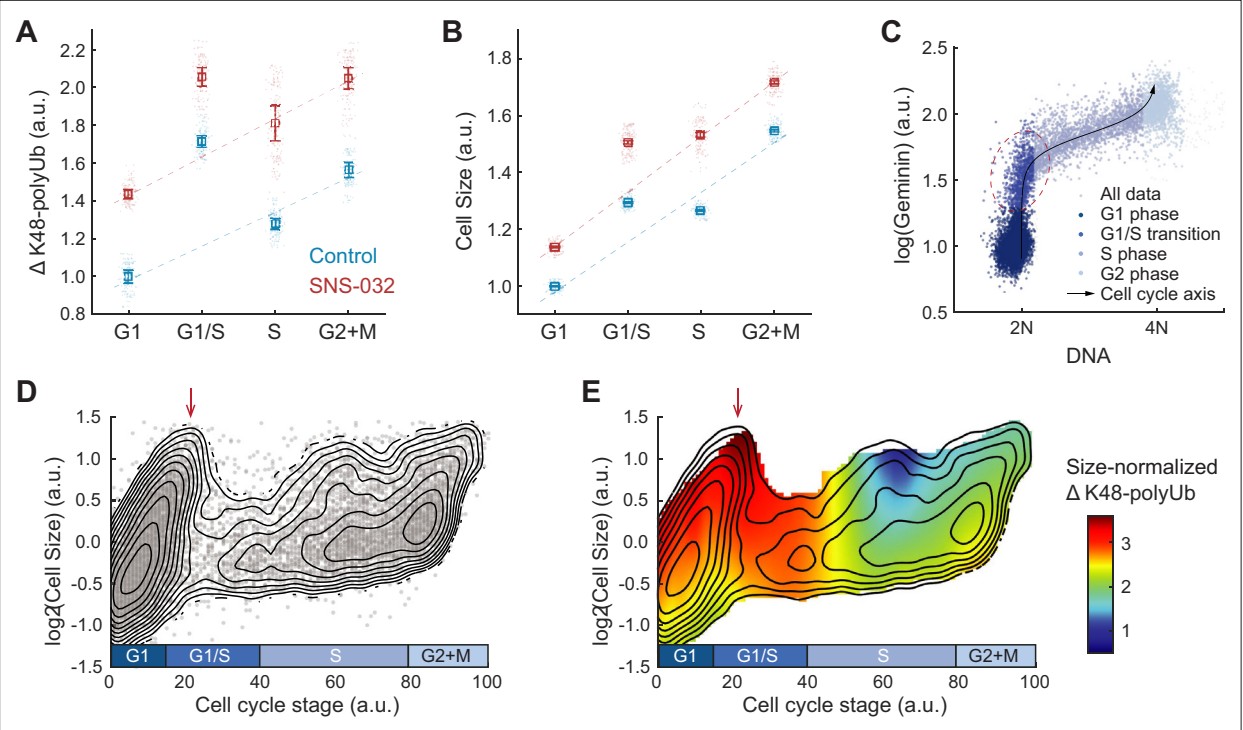

**Figure 4.** Hyperactivation of global protein degradation in naturally large cells at the G1/S transition. (**A, B**) ΔK48-polyUb (**A**) and cell size (SE, **B**) for control (0.1% v/v DMSO) and cells treated with SNS-032 (25 nM, 24 hr), separated by cell cycle stage. Data presented as median ± 95% CI, N = 9 replicate wells. Lines show linear fit, excluding G1/S data, which deviates from the trend of G1, S, and G2+M cells. (**C**) Single-cell measurements of DNA content (DAPI) and the cell cycle reporter Geminin (mAG-hGem) in an unsynchronized population of proliferating RPE1 cells. Black curve represents an average cell cycle trajectory, which is used to parameterize the progression through the cell cycle into a 1D curve used in panels (**D, E**) (see 'Materials and methods'). Red circle highlights the G1/S cells, which are large in size and high in ΔK48-polyUb. (**D**) Single-cell measurements of cell size (shown in log scale) as a function of the cell cycle trajectory (see **C** and 'Materials and methods') demonstrated in a scatterplot, overlaid with contour lines representing the calculated joint probability density function, which describes the frequency of cells for every given paired value of cell size and cell cycle stage. (**E**) Heatmap of size-normalized ΔK48-polyUb (see 'Materials and methods') overlaid on the density contours as in (**D**). Red arrows point to the large cells at G1/S transition. See **Figure 4—figure supplement 1** for heatmap of non-size-normalized ΔK48-polyUb.

The online version of this article includes the following source data and figure supplement(s) for figure 4:

**Source data 1.** File contains the source code and source data necessary to generate **Figure 4**, **Figure 4—figure supplement 1** using MATLAB.

**Figure supplement 1.** Large cells prior to the G1/S transition continue through the cell cycle and show elevated ΔK48-polyUb.

**Figure supplement 2.** Large cells prior to the G1/S transition show hyperactivated global protein degradation as demonstrated by live-cell imaging.

**Figure supplement 2—source data 1.** File contains the source code and source data necessary to generate **Figure 4—figure supplement 2** using Matlab.

## Discussion

Cell size is fundamental to cellular physiology as it sets the scale for subcellular compartments, biosynthetic capacity, and cellular function. Evidence of cell size control has been reported from single-celled yeasts to multicellular animals and plants (**Dolznig et al., 2004**; **Hartwell et al., 1974**; **Neufeld et al., 1998**; **Nurse, 1975**; **Willis et al., 2016**; **Xie and Skotheim, 2020**; **Zetterberg and Killander, 1965**). The precision with which size is controlled also manifests in the cell size regularity observed in healthy tissues. In contrast, deregulation of size control often signals cancerous growth as many tumors display increased heterogeneity in cell size (**Asadullah et al., 2021**; **Bell and Waizbard, 1986**; **Lee et al., 1992**).

Previous studies on cell size control have predominantly focused on cell size checkpoints, in that a critical size threshold is required for cell cycle progression (**D'Ario et al., 2021**; **Ginzberg et al., 2018**; **Liu et al., 2018**; **Sellam et al., 2019**; **Zatulovskiy et al., 2020**). In this study, we investigated size-dependent compensatory growth, which functions independently of cell cycle checkpoints. We employed a CDK2 inhibition assay to trigger compensatory growth, forcing cells to undergo

prolonged growth periods, resulting in an initial increase in cell size followed by a delayed compensatory growth slowdown (*Figure 1*). We found that the rates of global protein synthesis and degradation increase with cell size in both perturbed and unperturbed conditions (*Figures 2 and 3*). Interestingly, although protein synthesis rates scaled proportionally with cell size, protein degradation rates were disproportionately higher in large CDK2-inhibited cells (i.e., higher rates of protein degradation per unit mass), suggesting active upregulation of the proteasomal degradation pathway (*Figure 3*). In contrast, CDK4/6 inhibition, which induces a larger target size without triggering compensatory growth slowdown, exhibited the same linear scaling between cell size and protein degradation rates as in control (*Figure 3—figure supplement 4*). We also tested two other perturbations that forced overgrowth in cell size by extending S or G2/M phases. Like CDK2 inhibition, these perturbations induced superlinear increases in protein degradation (*Figure 3—figure supplement 4*), suggesting that activation of protein degradation functions as a general compensatory mechanism of cell size control in response to prolonged growth periods, irrespective of the specific cell cycle stage affected. Further analysis across the cell cycle highlighted a particularly striking increase in global protein degradation in large cells at the G1/S transition (*Figure 4*), consistent with the stringent size control observed at the G1 exit (*Ginzberg et al., 2018*; *Zetterberg and Killander, 1965*). Based on these results, we propose a model in which oversized cells reduce their growth efficiency by activating global proteasome-mediated protein degradation to promote cell size homeostasis (*Figure 5*).

While this study focused on human RPE1 cells, a non-cancerous epithelial cell line, previous work (*Mu et al., 2020*) on a cancerous mouse lymphocytic leukemia cell line found that large polyploid cells do not reduce growth efficiency compared to smaller diploid cells, suggesting that growth rate regulation may be cell type- and ploidy-dependent. Previous work has also suggested that small cells may accelerate their growth at certain cell cycle stages (*Cadart et al., 2018*; *Ginzberg et al., 2018*). It will be interesting to examine in future studies whether the growth acceleration observed in these small cells depends on reduced global protein degradation. High-throughput screens on both animal cells and yeasts identified that small cells activate the stress-responding p38 MAPK (Hog1) pathway to prolong the cell cycle (*Liu et al., 2018*; *Sellam et al., 2019*). Previous work on p38 also found that activation of the pathway by osmotic stress results in reduced protein degradation and lower proteasome activity (*Lee et al., 2010*), whereas p38 inhibition significantly promoted proteasome activity (*Leestemaker et al., 2017*). These reports suggest an intriguing possibility that the p38 MAPK activation in small cells may function in both the cell size checkpoint and growth rate regulation.

Our data also suggest hyperactivated rates of global protein degradation in large cells at the G1/S transition (*Figure 4*). This suggests that, in addition to accommodating cell size checkpoints, the G1/S transition may also serve as a critical regulatory point for protein degradation and growth rate control. These findings align with previous observation of decreased variability in both cell size and growth rate at the G1/S transition (*Ginzberg et al., 2018*; *Kafri et al., 2013*; *Son et al., 2012*; *Zetterberg and Killander, 1965*). It is well established that progression through the cell cycle, including the G1/S and G2/M transitions, is controlled by ubiquitin-mediated degradation of cell cycle regulators (*Barr et al., 2016*; *Nakayama and Nakayama, 2006*). It will be important in future studies to identify the molecular mechanisms underlying the compensatory protein degradation in large cells. For example, what are the enzymes responsible for the observed increase in K48-linked polyubiquitination in larger cells during the G1/S transition? Are compensatory degradation and cyclin turnover mediated by the same, or separate, factors?

In this work, we identified a size-dependent regulation of global protein degradation that contributes significantly to proteasome-mediated degradation. We also found that both nascent proteins and long-lived proteins are targeted for degradation during compensatory growth slowdown, with the relative contribution of each depending on the cell cycle stage (*Figure 3G*, *Figure 3—figure supplement 6*). While proteasomal degradation of nascent chains is classically associated with protein quality control, our data suggest that cells leverage this pathway to modulate their growth rate and maintain cell size homeostasis. Future work employing dynamic isotopic labeling and quantitative proteomics could provide valuable insights into the degradation rates of specific proteins in this context. Two recent studies *Cheng et al., 2021*; *Lanz et al., 2021* used such an approach and measured individual protein concentrations in cells of different sizes. Proteasome subunits were found at higher concentrations in large cells, consistent with elevated rates of ubiquitylation and protein turnover. Interestingly, components involved in translation show slightly reduced concentrations in large cells. Previous

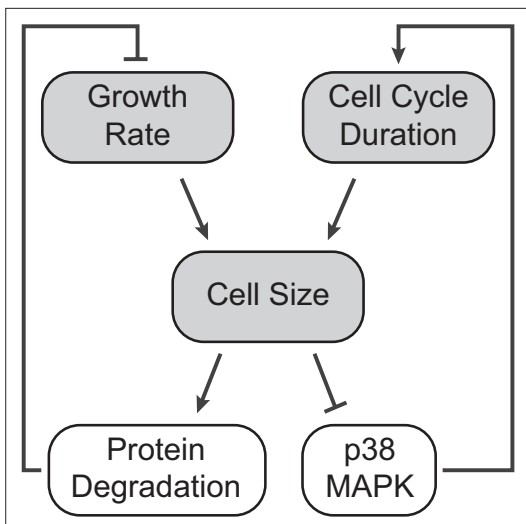

**Figure 5.** A model of cell size homeostatic control. Control of cell size homeostasis involves negative feedback on both the cell cycle duration and cellular growth rate. The cell cycle arm of the regulation involves activation of the p38 MAPK pathway in small cells that prolongs the growth duration (*Liu et al., 2018*). The growth rate arm of the regulation, investigated here, involves upregulated global protein degradation through the proteasomal degradation pathway that reduces growth efficiency in large cells.

work in yeast *Neurohr et al., 2019* demonstrated that oversized cells show impaired gene expression. If translation integrity is disrupted in large cells, increased levels of ubiquitin-mediated degradation may be required for the clearance of aberrant nascent polypeptides through protein quality control pathways. This is consistent with our finding that nascent proteins can significantly contribute to compensatory degradation.

Our findings provide mechanistic insight into the process of compensatory growth which, unlike cell size checkpoints, can be employed in the homeostatic size control of nonproliferating cells. This has important implications for understanding how terminally differentiated cells, which constitute the majority of cells in an adult body, maintain their size and function. Maintaining cells within their optimal size range is crucial for cellular and organ function. In adipocytes, the release of fatty acid induced by epinephrine was found to be highly dependent on the cell surface area (*Zinder and Shapiro, 1971*). Pancreatic beta cells undergoing hypertrophy show enhanced insulin secretion and improved glucose homeostasis in vivo (*Helman et al., 2016*). Across tissue types, cells dynamically control their size to modulate cell- and tissue-level functions in response to physiological demand; these include compensatory hypertrophy in pancreatic and hepatic cells (*Dhawan et al., 2007*; *Ginzberg et al., 2015*; *Miettinen et al., 2014*), compensatory renal cell hypertrophy following nephrectomy (*Rojas-Canales et al., 2019*) or increased urine flow (*Boehlke et al., 2010*), growth of neurons (axons) in coordination with animal development (*Albus et al., 2013*), and the likely regulation of functional mass of the hormone-secreting glands in response to stress (*Karin et al., 2020*; *Karin et al., 2021*). These examples highlight the importance of cell size control in nonproliferating, differentiated cells. The experimental pipeline used in this study can be adapted to study the roles of protein synthesis and degradation in cell size control of nonproliferating cells, whether certain mechanisms are shared between proliferating and nonproliferating cells, and how these processes affect cellular functions.

## Materials and methods

This study did not generate new unique reagents. All compounds were purchased from Selleckchem (SNS-032, S1145; palbociclib, S4482; bortezomib, S1013; cycloheximide, S7418; carfilzomib S2853; JNJ-26854165, S1172; barasertib, S1147). Click-IT L-azidohomoalaine was purchased from Thermo Scientific (C10102). The activity-based active proteasome probe (ABP) MV151 Bodipy-TMR was a kind gift from the Florea Lab at Leiden University. The anti-ubiquitin K48-specific antibody (Clone Apu2.07) was a kind gift from Genentech.

### Cell culture

This work used human retinal pigmented epithelial (RPE1) obtained from ATCC (RRID:4388). Cells have been regularly tested for mycoplasma contamination, and no contamination had been detected during the experiments. RPE1 cells with stable expression of the degron of Geminin fused to Azami Green (mAG-hGem) were cultured in DMEM medium (Life Technologies) supplemented with 10% fetal bovine serum (FBS, Wisent, Montreal, QC) at 37°C in a humidified atmosphere with 5% $CO_2$.

Measurements were generally made when cells were 60–80% confluent to avoid the effects of sparse or dense culture on cell growth and proliferation.

## Fixation, staining, and imaging

Cells were fixed in 4% paraformaldehyde (Electron Microscopy Sciences, Hatfield, PA) for 10 min, followed by permeabilization in cold methanol at –20°C for 5 min. Cells were stained with 0.4 μg/mL Alexa Fluor 647 carboxylic acid, succinimidyl ester (SE-A647, Invitrogen A-20006) for 2 hr at room temperature. The cellular integrated intensity of SE-A647 measures total protein mass, which is proportional to a cell's dry mass (*Kafri et al., 2013*; *Mugahid et al., 2020*). DNA was stained with 1 μg/mL DAPI (Sigma D8417) for 10 min at room temperature. Cells were imaged using the Operetta High-Content Imaging System (PerkinElmer, Woodbridge, ON) at ×20 magnification. Automated image processing was performed as previously described (*Liu et al., 2018*).

## Time-lapse microscopy and analysis

RPE1 cells with stable expression of H2B-mTurquoise and Geminin-mVenus were seeded in 96-well μclear microplates (Greiner Bio-one, Monroe, NC) and grown in the incubator for at least 6 hr prior to imaging. The cells were imaged using the Operetta High-Content Imaging System. During imaging, the plate was incubated in a live-cell chamber (37°C, 5% $CO_2$), and cells were grown in FluoroBrite DMEM supplemented with FBS, L-glutamine and sodium pyruvate. As the cells were previously cultured in regular DMEM and displayed suboptimal cell proliferation after switching to FluoroBrite DMEM, cells were grown in FluoroBrite medium for a period of 2 weeks to adapt to the new medium before the time-lapse experiments. Widefield fluorescent images of H2B-mTurquoise and Geminin-mVenus were collected every 15 min at ×20 magnification for 60 hr. Under this experimental setting, the microscope could support imaging of up to four wells. To track the movement and division of single cells, and analyze nuclear area dynamics and cell cycle progression, we used the same methods as described previously (*Liu et al., 2018*). Growth rate is estimated as the first derivative of the smoothed nuclear area dynamics, and all measurements presented in the study only examined the cells continuously tracked from one division to the next. The first and last six time points of the cell cycle were removed from the growth rate analysis because of the influence of nuclear breakdown and formation.

## Cell cycle stages

Cells were first partitioned, according to their integrated nuclear DNA level, into G1 (2N), S (2N-4N), and G2 (4N) phases. Progression through the G1 phase was further divided, based on the fluorescence of integrated nuclear Geminin, when available, into early G1 (low baseline Geminin) and G1/S transition (higher Geminin). The thresholds were automatically detected based on the distributions of DNA and log(Geminin).

## Measures of AHA incorporation

To quantify rates of nascent protein synthesis, cells were treated with the CDK4/6 inhibitor palbociclib (50 nM) or CDK2 inhibitor SNS-032 (25 nM) for 48 hr, then pulse-labeled with Click-IT L-azidohomoalanine (AHA, Invitrogen C10102) for 3 hr as described in the manufacturer's protocol. Cells were then fixed and stained for DAPI and SE-A647. Rates of AHA incorporation were determined by labeling the cells with Alexa Fluor 488 DIBO alkyne (Invitrogen C10405), followed by imaging and quantification of the integrated intensity of the fluorophore on a single-cell basis as detailed above. As a negative control, cells were treated with 1 μM of the protein synthesis inhibitor CHX.

## Cycloheximide chase experiment

Cells were seeded at 1500 cells per well into 96-well Cell Carrier-96 ultra microplates (PerkinElmer) for at least 6 hr prior to treatment. The protein synthesis inhibitor CHX was then administered at 10 μM for either 0, 1.5, 3, 6, or 9 hr. Cells were then fixed and stained for DAPI and SE-A647, and imaged as detailed above. Total protein loss over time was measured by changes in cell size (SE) for the smaller 20% (20th percentile) and the larger 20% (80th percentile) of G1, S, or G2 cells separately.

## Measurements of active proteasomes

RPE1-mAG-hGem cells were seeded at 1500 cells per well into 96-well Cell Carrier-96 ultra microplates (PerkinElmer) for at least 6 hr prior to treatment. The cells were then treated with 1 μM activity-based

active proteasome probe (ABP) MV151 Bodipy-TMR for 2 hr. MV151 binds to the inside of the catalytic core (20S) of active proteasomes, providing a total fluorescence intensity (per cell) that is proportional to proteasomal activity (*Verdoes et al., 2006*). As a negative control, cells were treated with 1 µM of the proteasome inhibitor bortezomib. Cells were then fixed and stained for DAPI and SE-A647, and imaged as detailed above.

## Measurements of K48-polyUb turnover

RPE1-mAG-hGem cells were seeded into 96-well Cell Carrier-96 ultra microplates (PerkinElmer) for at least 6 hr prior to treatment. Cells were then treated with the experimental drugs (e.g., 25 nM SNS-032) or DMSO control (<0.5% v/v) for the intended treatment time as indicated in the figures. At 30 min before fixation, half of the wells of each experimental condition were treated with a proteasome inhibitor (8 µM CFZ). After fixation, cells were immunostained for total levels of K48-linked polyubiquitin with a primary antibody against K48-polyUb chains (Clone Apu2.07, Genentech, 1:500) for 2 hr at room temperature, followed by incubation with a secondary antibody (goat anti-human IgG (H+L) Cross-Adsorbed Secondary Antibody, Alexa 555, Thermo Fisher, 1:500) for 30 min at room temperature. Cells were then stained for DAPI and SE-A647, and imaged as detailed above. ΔK48-polyUb is calculated between every pair of CFZ-treated vs. non-CFZ-treated wells of the same experimental condition. Average and CIs of ΔK48-polyUb were then calculated with all replicate measurements of the same condition.

## Whole-cell lysis and western blotting

To prepare whole-cell lysates, cells were rinsed with ice-cold PBS and solubilized with RIPA Lysis Buffer (Boston Bio-Products, Boston, MA) (50 mM Tris-HCl, 150 mM NaCl, 5 mM EDTA, 1 mM EGTA, 1% NP-40, 0.1% SDS and 0.5% sodium deoxycholate, pH 7.4) supplemented with protease and phosphatase inhibitor cocktail (Thermo Scientific, Burlington, ON). Protein concentration was determined using the BCA protein assay (Thermo Scientific). Proteins were suspended with 4× Bolt LDS Sample Buffer and 10× Bolt Reducing Agent and heated for 10 min at 70°C. Samples of equal protein were resolved by SDS-polyacrylamide gel electrophoresis and subjected to immunoblotting for proteins as indicated. All western blot results in the figures have been reproduced in replicate experiments with cell lysates samples prepared in independent experiments.

## Estimation of cell cycle length and growth rate from bulk measurements

Cells were treated with inhibitors on multiple 96-well plates and fixed every 20 hr over a period of 3 days. The plate slated to be fixed on the last timepoint was imaged by digital phase contrast (Operetta High-Content Imaging System; PerkinElmer) every 12 hr to acquire cell number estimates. Cell size was quantified using the total fluorescence intensity from SE-A647 at a single-cell level. Growth rate and cell cycle length were quantified from population averages of cell size and cell number over time. To quantify cell cycle length ($\tau$), we fit exponential curves to cell number over time ($N_t = N_o e^{\alpha t}$), where $N_t$ is the cell count at time $t$ and $\alpha = \frac{ln(2)}{t}$. To estimate growth rate ($\nu$), we calculated the rate of increase in bulk mass ($M_t = cell\ size \times cell\ count$) of the total population and divided that by the cell number: $\nu = \frac{1}{N_t}\frac{dM}{dt}$. Due to our method of cell size measurement, growth rate quantifications were performed on fixed populations of cells.

## Estimation of a continuous cell cycle axis

Single-cell levels of DNA (DAPI) and Geminin (mAG-hGem) were reduced to a single variable ($\ell$), which represents a continuous measure of cell cycle stages. See specific algorithm described previously (*Kafri et al., 2013*). In brief, the trajectory is detected as the probability density ridge in the DNA-Geminin distribution, and individual cells were projected to the trajectory through the shortest distance. Note the cell cycle trajectory detected in this study corresponds to an average progression over the cell cycle but does not necessarily reflect the relative duration of each phase.

## Acknowledgements

We thank Yifat Merbl, Mikael Björklund, and members of the Kafri laboratories for helpful discussions. This work was supported by the Natural Sciences and Engineering Research Council of Canada (RGPIN-2015-05805) to RK, the Restracomp Graduate and Postdoctoral Fellowship from the Research Training Center at the Hospital for Sick Children to SL and MBG, the University of Toronto Open Fellowship to CT, and the National Institute of General Medical Sciences of the National Institutes of Health award F32GM120956 to KGM.

## Additional information

### Competing interests

Michael Rape: Reviewing Editor eLife, founder and member of the scientific advisory board of Nurix Therapeutics, a member of the scientific advisory board of Monte Rosa Therapeutics, and an iPartner with The Column Group. The other authors declare that no competing interests exist.

### Funding

| Funder | Grant reference number | Author |
|---|---|---|
| Natural Sciences and Engineering Research Council of Canada | RGPIN-2015-05805 | Ran Kafri |
| Hospital for Sick Children | Restracomp Graduate Fellowship | Shixuan Liu |
| Hospital for Sick Children | Restracomp Postdoc Fellowship | Miriam B Ginzberg |
| University of Toronto | Open Fellowship | Ceryl Tan |
| National Institute of General Medical Sciences | F32GM120956 | Kevin G Mark |

The funders had no role in study design, data collection and interpretation, or the decision to submit the work for publication.

### Author contributions

Shixuan Liu, Conceptualization, Data curation, Formal analysis, Supervision, Validation, Investigation, Visualization, Methodology, Writing – original draft, Project administration, Writing – review and editing; Ceryl Tan, Data curation, Formal analysis, Validation, Investigation, Visualization, Methodology, Writing – original draft, Project administration, Writing – review and editing; Chloe Melo-Gavin, Miriam B Ginzberg, Data curation, Formal analysis, Validation, Investigation, Methodology, Writing – review and editing; Ron Blutrich, Investigation, Methodology; Nish Patel, Supervision, Investigation, Methodology, Project administration, Writing – review and editing; Michael Rape, Resources, Supervision, Investigation, Methodology, Project administration, Writing – review and editing; Kevin G Mark, Resources, Data curation, Supervision, Validation, Investigation, Methodology, Writing – review and editing; Ran Kafri, Conceptualization, Resources, Formal analysis, Supervision, Funding acquisition, Investigation, Visualization, Writing – original draft, Project administration, Writing – review and editing

### Author ORCIDs

Shixuan Liu ⓘ https://orcid.org/0000-0003-4972-415X
Ceryl Tan ⓘ https://orcid.org/0000-0002-9010-9039
Michael Rape ⓘ https://orcid.org/0000-0003-4849-6343
Ran Kafri ⓘ https://orcid.org/0000-0002-9656-0189

### Decision letter and Author response

Decision letter https://doi.org/10.7554/eLife.75393.sa1
Author response https://doi.org/10.7554/eLife.75393.sa2

# Additional files

## Supplementary files
Transparent reporting form

## Data availability
All data presented in this study are included in the manuscript and supporting files. Source data files have been provided for all figures.

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
