## [Editor Report]

This important study reports a previously undocumented role for ubiquitin-proteasome system (UPS)-mediated protein turnover in size control in human cells. The authors show that large cells undergo size compensation by actively reducing their rate of growth and this effect is shown to be mediated by an increase in the rate of proteasome-mediated degradation. The experiments are well controlled, and the conclusions of the study are compelling and well supported by the data. Overall the paper increases our knowledge related to size control mechanisms in dividing and non-dividing cells.

---

## [Decision Letter]

**Decision letter after peer review:**

Thank you for submitting your article "Large cells activate global protein degradation to maintain cell size homeostasis" for consideration by *eLife*. Your article has been reviewed by 3 peer , including Bruce A Edgar as the Reviewing Editor and Reviewer #1, and the evaluation has been overseen by Marianne Bronner as the Senior Editor. We apologize for the very long time this manuscript has taken in review.

Overall, the reviewers appreciated the significance of the new central finding, that protein degradation can be used as a mechanism to regulate cell size, and compensate for changes in division rates. However, for several reasons, the reviewers were not entirely confident in the results used to support this interesting mechanism, and so they have requested major revisions with new data and new data analyses.

Essential revisions (for the authors):

1) As reviewer 2 noted: "something is wrong with the cell size measurements in Figure 2 because many cells basically have almost negligible size (near 0) while others have sizes up to 5 or 6 arbitrary units. It makes no sense that there should be a 10-fold or even 100-fold range in cell sizes… The data supporting higher rates of protein degradation per unit mass in large cells suffers from a similar problem as Figure 3E has the same issue as Figure 2 with too many tiny 'cells'." Please address this concern with an explanation, and if necessary, add new data or a new analysis of existing data.

2) Review #1 was concerned that only CDK2 inhibition was used as a tool to alter rates of cell cycle progression. The reviewer reasonably requests that an alternate tool be used to demonstrate that the effect on protein degradation is a general one, and not a direct and specific effect of CDK2 inhibition.

3) Reviewer #3 makes the important point (2) that "If translation rate is higher in large cells, then basal protein degradation may also be correspondingly higher." Some control experiments should be provided that rule out this trivial possibility.

4) The reviews mention many other small and large changes that could improve the paper. Please consider these and address them at your discretion.

*Reviewer #1 (Recommendations for the authors):*

The paper falls short of demonstrating, or proposing, a molecular mechanism explaining the connection between cell size and protein degradation. This is not strictly necessary provided the basic phenomenon is definitely demonstrated, but additional data on mechanism would certainly elevate the paper considerably.

1. A very relevant paper on cell growth from Mu, Meittinen et al. (PNAS 2020) should be discussed and cited. This paper concludes that larger cells (of a different type) do not grow slower.

2. The figure legends generally have too little information about the experimental design, making it difficult to interpret the results. In some cases the relevant details about an experiment can be found in the methods, but not always. Please review the figure legends and add detail explaining how the experiment was done.

3. Figure 1 C presents "early" and "late" cells. Please denote the time/treatments these cells received.

4. Figure 3A is not really informative, since the change in cell mass is shown in AU rather than % of cell mass. Please show this as a fraction of cell mass.

5. Figure 3E is essentially a control example, but it is presented as a result. To support the hypothesis that this data shows more active proteasomes in larger cells, the data should: (a) include a similar dataset for SNS-032 treated cells; and (b) be normalized to cell sizes. From the data presented, it looks like active proteasome amounts scale linearly with cell size, and that larger cells do not have a fractionally high number of active proteasomes (per unit mass). This doesn't support the authors' hypothesis.

6. The authors' general conclusions might be tested by decreasing protein degradation or ubiquitination in the CDK2-inhibited cells. This would be predicted to further increase cell size.

7. Line 467-468: were the deltaK48-polyUB data in Figure 4 supplement 1 also normalized to cell volumes? Please clarify in the figure legend and/or methods.

8. The authors present a number of assays with immunodetection of K48-polyUB. I believe what they are measuring here is k48-polyubiquitinated proteins of all sorts. If so, they should refer to this metric as "K48-polyUB-protiens" in the text and figures.

9. Figure 3F needs to present statistical differences between the SNS-032-treated and control cells.

10. It was not clear to me how Figure 4 supplement 2 supports the conclusions of the paper. The authors should explain this better, or leave it out.

11. It is interesting that generally cells with larger size have higher protein degradation rate (S, G2 cells degrade proteins faster than early G1), except that G1/S cells possess the highest degradation rate among all cell cycle phases. Since G1/S cells are not in the phase with the largest cell size, this is rather puzzling. Does it suggest that protein degradation rate is primarily regulated by some other mechanism, like CDK activity, rather than cell size? Please comment.

12. In Figure 1B, the authors characterize the average cell size changes after rapamycin or SNS-032 treatment. Because cell size is a distribution over a range, simply averaging all cells together may obscure some informational content. It is probably preferable to plot the distribution of cell sizes (e.g. dot plots or violin plots), comparing control and treatment groups.

13. Figure 1D shows that the nuclear area drops around G1/S when Cdk2 inhibitor is used. However, when they replotted the data by aligning cells based on the timeline relative to G1/S, this phenotype is not obvious anymore (Figure 1E). Please comment on why this is.

14. Page 3, line 112: should the reference be to Figure 1B?

*Reviewer #3 (Recommendations for the authors):*

1. While the main conclusion of the paper that larger cells size compensate by increased proteasome-dependent protein turnover is supported by key experiments (Figure 3 D,E; Figure 4 A,E) the link to cell size is largely correlative, as supported by exploiting stochastic size variation in unperturbed cells and treatment with the CDK2 inhibitor in a perturbed context. To my mind, an important control would be to look in the situation where cells become large and size compensation does not occur, i.e., CDK6 inhibitor treated cells. The authors' model predicts that there should be no increase in proteasomal activity in the CDK6 inhibitor-treated cells. It seems likely that authors would have considered this experiment given their published observation that CDK6 inhibition is accompanied by unrestrained growth without size compensation.

2. One mechanism that could contribute to the increased rate of degradation is abortive translation that requires proteasome mediated clearance of partial translation products. It has long been known that turnover of nascent translated proteins accounts for a substantial fraction of proteasomal flux (~50% if I recall correctly). If translation rate is higher in large cells, then basal protein degradation may also be correspondingly higher. Although the size normalization of K48 chain turnover suggests that additional degradation indeed occurs in large cells, a possible further control to test this idea would be to treat cells with an acute sub-lethal dose of cycloheximide to slow translation followed by measurement of proteasome activity. Alternatively, treatment of cells with a low dose proteasome inhibitor would be predicted to reduce size compensation and allow cells to become larger.

---

## [Author Response]

Essential revisions (for the authors):1) As reviewer 2 noted: "something is wrong with the cell size measurements in Figure 2 because many cells basically have almost negligible size (near 0) while others have sizes up to 5 or 6 arbitrary units. It makes no sense that there should be a 10-fold or even 100-fold range in cell sizes…… The data supporting higher rates of protein degradation per unit mass in large cells suffers from a similar problem as Figure 3E has the same issue as Figure 2 with too many tiny 'cells'." Please address this concern with an explanation, and if necessary, add new data or a new analysis of existing data.

The reviewer is correct that the original scatter plots in Figure 2 gave a misleading impression of the cell size distribution, suggesting an implausible 10-fold to 100-fold variation in cell sizes – a false impression that was unfortunately caused by our poor choice of data representation. The previous visualization inadvertently emphasized the presence of rare image processing cell-segmentation errors that make the appearance of extremely-sized “cells”. Although these outliers are rare in our scatter plots, they are visually prominent, leading to an overestimation of the actual cell size range. It is worth noting that when processing thousands of images to extract single cell data, the presence of such rare segmentation errors is currently inevitable. While these false positives do not affect statistics nor overall trends, they do jump to eye when presented as scatter plots, especially in the sparse regions of the scatter plots.

To correct this misleading representation, we now revised Figure 2 to more effectively depict the actual cell size distributions in our data and de-emphasized the outliers laying at the extremes. The inclusion of cell size histograms further clarifies the true distribution of cell sizes. To rectify the misleading representation in the original manuscript, we have implemented the following changes:

Data Representation: We have applied a transparency setting to the scatter plot data points. This method offers a more accurate depiction of the data distribution.Cell Size Histograms: We have included histograms in Figure 2A to illustrate the actual distribution of cell size. This provides a clearer view of the range of our measurements.Normalization: We now present cell size data normalized to the average G1 size in control conditions, whenever applicable. This normalization helps to interpret cell size measurements within a more meaningful context, allowing for better comparison across different cell cycle stages and treatment conditions.

By implementing these changes, we aim to provide a more accurate representation of our data and address the reviewer's concerns effectively. These adjustments have also been applied to Figure 3E (currently Figure 3F) to ensure consistency and clarity across our analyses.

As clarified by the revised figure, our cell size measurements show a much narrower and biologically plausible range (see Author response image 1 and revised Figure 2A). For example, when comparing the average of G1 phase cells to that of the G2+M phase, we observe a ~1.5-fold change in size, which aligns with typical expectations.

**Author response image 1. sa2fig1:** Cell size distribution in DMSO control condition. A. Cell size histogram with median (solid line) and 1st and 99th percentiles (dashed lines) labeled. B. DNA histogram showing classification of cells to different cell cycle stages. C. Cell size distribution plotted separately for G1, S, and G2+M cells with their median labeled by solid lines of the corresponding color.

2) Review #1 was concerned that only CDK2 inhibition was used as a tool to alter rates of cell cycle progression. The reviewer reasonably requests that an alternate tool be used to demonstrate that the effect on protein degradation is a general one, and not a direct and specific effect of CDK2 inhibition.

We accept the reviewer’s critique. In the revision, we explored additional perturbations that slow rates of cell cycle progression, and these new data have further strengthened our conclusions.

Generally speaking, cell cycle lengthening can be caused by delaying progression through G1, S, or G2/M phases. With respect to G1, the three primary targets are CDK2, CDK4, and CDK6. However, in the context of the present study, only CDK2, but not CDK4/6, is an appropriate target. As described in our previous studies (Ginzberg et al., 2018, doi.org/10.7554/*eLife*.26957; Tan et al., 2021, doi.org/10.1016/j.devcel.2021.04.030), inhibiting CDK2 extends the duration of G1, leading to an initial increase in cell size that is subsequently counteracted by a compensatory slowdown in growth rate. In contrast, inhibiting CDK4/6 not only affects cell cycle progression but also alters the cell's target size. Thus, we applied CDK4/6 inhibition (by palbociclib) as an additional “negative control” in the revision. Indeed, we found that although CDK4/6 inhibition significantly increases cell size and protein degradation rates, the two maintain a similar linear relationship as the control (Figure 3—figure supplement 4A-C).

We also expanded our study and included in our revision not only G1 perturbations, but also perturbations that lengthen the cell cycle by slowing down progression through S phase and mitosis. We identified chemicals from a previous in-house screen (Kaufman et al., 2022, doi.org/10.1038/s41467-022-30008-0) that significantly altered cell size while extending the S phase or G2/M phase, without significantly disrupting cell size uniformity. Another criterion was to avoid perturbations that directly disrupt the structural integrity (e.g., DNA, cytoskeleton) of the cell. This allowed us to employ two alternate compounds: JNJ-26854165, which slows S phase progression, and barasertib, which inhibits mitosis. The new data (Figure 3—figure supplement 4D-I) revealed that inhibiting progression of either S or M phase triggered a compensatory increase in global protein degradation, as evidenced by elevated levels of ΔK48-polyUb. This effect is particularly pronounced in the specific cell cycle stages targeted by the inhibitors. And like CDK2 inhibition, these increases in ΔK48-polyUb are disproportionately larger than the corresponding increases in cell size, suggesting an active upregulation of protein degradation beyond what would be expected solely due to increased cell size. These new results suggest that the activation of protein degradation functions as a general compensatory mechanism for oversized cells to maintain size homeostasis, regardless of the specific cell cycle stage affected.

These new data and analyses have been incorporated in the revised manuscript line 224-253.

3) Reviewer #3 makes the important point (2) that "If translation rate is higher in large cells, then basal protein degradation may also be correspondingly higher." Some control experiments should be provided that rule out this trivial possibility.

The reviewer's insightful comment raises two important questions:

Is the higher rate of protein degradation in large cells simply due to larger cells having more proteins, rather than a compensatory mechanism?Which types of protein are degraded: newly synthesized nascent proteins (canonically associated with protein quality control) or non-nascent/long-lived proteins?

Addressing the first question: While we do observe a positive correlation between cell size and protein degradation rate, cells under CDK2 inhibition show a superlinear increase in ΔK48-polyUb relative to their increase in cell size. This is particularly evident when examining different cell cycle stages (Figure 3D). While control cells in G1, S, and G2+M phases show a positive correlation between size and ΔK48-polyUb, CDK2 inhibited cells sit above this trend, indicating an increase in protein degradation beyond what would be expected from increased cell size alone. In addition, naturally large cells at G1/S transition also show higher efficiency of protein degradation (i.e., degradation rates per unit mass) compared to similarly-sized or even larger cells in S and G2 (Figure 4E). These observations are not consistent with the simple model that large cells have higher protein degradation simply because of their higher translation rate. In the new manuscript, we now better present these points in Results line 203-253 and line 296-314, and in Discussion line 341-365.

To address the second question, we followed the reviewer's suggestion and measured proteasome-mediated protein degradation rates in cells subject to an acute, sub-lethal dose of cycloheximide (CHX, 3 μM for 3h). We applied this treatment with or without CDK2 inhibition (SNS+/-). This approach allowed us to discriminate degradation of nascent (CHX-sensitive) and non-nascent (CHX-insensitive) proteins:

ΔK48-polyUb in non-CHX-treated cells (CHX-) reflects degradation of both nascent and non-nascent proteinsΔK48-polyUb in CHX-treated cells (CHX+) reflects degradation of only non-nascent proteins Our results (Figure 3G, Figure 3—figure supplement 6) show:

ΔK48-polyUb significantly decreases with CHX treatment, consistent with previous understanding that nascent proteins constitute a substantial fraction of the proteasomal flux.In CHX+ conditions, SNS treatment significantly increases (+120%) degradation of non-nascent proteins in cells at the G1/S transition, but not at or to a much smaller extent at other cell cycle stages.SNS treatment increases ΔK48-polyUb across all cell cycle stages in CHX- conditions, indicating increased degradation of both nascent and non-nascent proteins, with significant contribution from nascent proteins.

These findings suggest that compensatory protein degradation involves both nascent and non-nascent proteins, with the balance depending on the cell cycle stage. Notably, this indicates that degradation of nascent proteins, typically associated with protein quality control, can be employed for growth rate modulation and control of cell size homeostasis. At the G1/S transition, a time point of stringent size control, large cells appear to activate degradation of non-nascent proteins to modulate growth rate. These new results are now incorporated in the revised manuscript line 278-295.

Regarding the reviewer’s suggestion to use a low-dose proteasome inhibitor, we have actually performed this very experiment in a previous publication (Kafri et al., 2013, doi: 10.1038/nature11897). Consistent with the thesis of the present study, low-dose proteasomal inhibition not only decreased cell size but also significantly increased cell size variance, particularly at the G1/S transition. Those results are consistent with our thesis that protein degradation serves to maintain cell size homeostasis.

In the current study, we used short-term (30 min) treatment with the proteasome inhibitor CFZ as a tool to quantify protein degradation rates. This short-term inhibition did not significantly affect cell size or cell cycle distribution (Figure 3—figure supplement 2). As for the suggestion for long-term proteasome inhibition, while the experiments can be interesting, we feel they may not be compatible with the goals and context of our present study. Because cell cycle progression is directly controlled by proteasome-mediated cyclin degradation, it would be challenging to distinguish between the inhibitor's effects on global protein metabolism, cell cycle perturbation, and cell size regulation.

4) The reviews mention many other small and large changes that could improve the paper. Please consider these and address them at your discretion.

We thank the reviewers for the additional suggestions. We respond to each point below.

Reviewer #1 (Recommendations for the authors):The paper falls short of demonstrating, or proposing, a molecular mechanism explaining the connection between cell size and protein degradation. This is not strictly necessary provided the basic phenomenon is definitely demonstrated, but additional data on mechanism would certainly elevate the paper considerably.

We share the reviewer’s curiosity in the molecular mechanisms connecting cell size and protein degradation. From years long research on the subject, we think the mechanisms in play should involve three distinct layers: (1) cell size sensing – biophysical or biochemical mechanisms that allow individual cells to detect their own size; (2) cell size-dependent signal transduction – signaling pathways downstream of the size sensors that mediate signaling events (e.g., gene expression, phosphorylation, etc.) depending on cell size; and (3) effector mechanisms – processes that regulate proteasomal activity based on the size-dependent signaling.

Identifying these mechanisms has been, and continues to be, a central focus of our lab and the field. However, while we are committed to pursuing this goal, addressing the full scope of these questions will require years of research that extends beyond the scope of the current paper. It is worth noting that despite the discovery of cell size checkpoints in the 1960s by Killander and Zetterberg, the molecular mechanisms of cell size sensors and checkpoints remain not well understood today.

In the revised manuscript, we have expanded the Discussion section to address the open questions raised by our study and highlight the importance of future research in elucidating the precise molecular mechanisms underlying this newly identified cell size control pathway (line 388-392, line 371-379).

1. A very relevant paper on cell growth from Mu, Meittinen et al. (PNAS 2020) should be discussed and cited. This paper concludes that larger cells (of a different type) do not grow slower.

We thank the reviewer for pointing us to this very relevant paper. We now cite and discuss the paper in the revised manuscript (line 366-369).

2. The figure legends generally have too little information about the experimental design, making it difficult to interpret the results. In some cases the relevant details about an experiment can be found in the methods, but not always. Please review the figure legends and add detail explaining how the experiment was done.

We have revised figure legends and added more details of the experimental design.

3. Figure 1 C presents "early" and "late" cells. Please denote the time/treatments these cells received.

We have now defined early and late stage cells in the respective legend (currently Figure 1 – S1).

4. Figure 3A is not really informative, since the change in cell mass is shown in AU rather than % of cell mass. Please show this as a fraction of cell mass.

We agree with the reviewer that showing changes as a fraction of cell mass is more informative, and have modified the figures (Figure 3A, Figure 3—figure supplement 1) accordingly.

5. Figure 3E is essentially a control example, but it is presented as a result. To support the hypothesis that this data shows more active proteasomes in larger cells, the data should: (a) include a similar dataset for SNS-032 treated cells; and (b) be normalized to cell sizes. From the data presented, it looks like active proteasome amounts scale linearly with cell size, and that larger cells do not have a fractionally high number of active proteasomes (per unit mass). This doesn't support the authors' hypothesis.

We agree with the reviewer that the linear correlation of cell size and active proteasomes does not fully explain the mechanism for size-dependent compensatory degradation. However, as emphasized by the reviewer comment, our description in the original manuscript failed to clarify that.

To explain the higher efficiency of protein degradation in oversized cells, the relationship between cell size and proteasomal degradation may follow three simple possibilities:

Oversized cells have higher concentrations of total proteasomes.Oversized cells have higher concentrations of active proteasomes.Oversized cells have higher efficiency in degrading proteins per active proteasome.

While there is superlinear increased degradation and polyubiquitination in larger cells, we clarified in the revised manuscript that large cells did not show superlinear increases at the level of total proteasome content or active proteasomes (Figure 3E-F). These data are more consistent with the third possibility of an increased efficiency in protein degradation, and we make it clear in the manuscript that the question warrants further investigation (line 272-277).

6. The authors' general conclusions might be tested by decreasing protein degradation or ubiquitination in the CDK2-inhibited cells. This would be predicted to further increase cell size.

We thank the reviewer for the suggestion. A related experiment has been previously reported, where we measured cell size dynamics under the proteasome inhibitor MG132 (Kafri et al., 2013, doi: 10.1038/nature11897). This study revealed that proteasome inhibition not only caused cells to become larger but also increased cell size variability, particularly at the G1/S transition. These findings align with our proposed model, suggesting that compensatory degradation acts as a general mechanism to buffer natural cell size variations.

For the current study, we considered applying long-term proteasome inhibition (e.g., 24 hours) in combination with CDK2 inhibitors. However, since cell cycle progression is tightly regulated by proteasome-mediated degradation of key cell cycle regulators, we believe that long-term inhibition would complicate interpretation. Specifically, it would be difficult to distinguish whether observed changes in cell size were due to global alterations in protein metabolism or direct disruptions to cell cycle regulation. For this reason, we opted not to pursue long-term proteasome inhibition in this context.

7. Line 467-468: were the deltaK48-polyUB data in Figure 4 supplement 1 also normalized to cell volumes? Please clarify in the figure legend and/or methods.

The ΔK48-polyUb data shown in Figure 4—figure supplement 1B are not normalized to cell size or volume. We now better clarify this in its figure legend (line 981) and in the results (line 311-312).

6. The authors present a number of assays with immunodetection of K48-polyUB. I believe what they are measuring here is k48-polyubiquitinated proteins of all sorts. If so, they should refer to this metric as "K48-polyUB-protiens" in the text and figures.

The reviewer is correct. We use “K48-polyUB” as an abbreviation for K48-linked polyubiquitinated proteins. We define this abbreviation when it first appears in the text (line 178-179) and in figure legends.

6. Figure 3F needs to present statistical differences between the SNS-032-treated and control cells.

As suggested, we have now included statistical significance for comparisons between SNS-032treated and control cells for each cell cycle stage in the figure legend (currently Figure 3D legend).

7. It was not clear to me how Figure 4 supplement 2 supports the conclusions of the paper. The authors should explain this better, or leave it out.

We appreciate the reviewer’s comment and understand the need for clarity.

The data presented in Figure 4—figure supplement 2A-B are derived from live-cell tracking experiments under mTOR inhibition. The key insight offered by the figure extends from the contrast between panels A and C. In panel A, mTORinhibited large and small cells show converging growth rates prior to the G1/S transition. However, as shown in panel C, mTOR-inhibited large cells synthesize proteins at a much faster rate than smaller cells. This discrepancy indirectly points to an elevated level of protein degradation in large cells prior to the G1/S transition. This result corroborates the more direct measurements of protein degradation presented in the main Figure 4.

In the revised manuscript, we have clarified the relevant Results section to better explain these points (line 315327). While reviewer #3 suggested including these data in the main figure, we believe it is more appropriate to keep it as a supplement to avoid overcomplicating the main figure layout.

8. It is interesting that generally cells with larger size have higher protein degradation rate (S, G2 cells degrade proteins faster than early G1), except that G1/S cells possess the highest degradation rate among all cell cycle phases. Since G1/S cells are not in the phase with the largest cell size, this is rather puzzling. Does it suggest that protein degradation rate is primarily regulated by some other mechanism, like CDK activity, rather than cell size? Please comment.

We thank the reviewer for pointing out this interesting phenomenon. We were indeed surprised to find that G1/S cells in control show the highest global protein degradation rates despite being generally smaller than S and G2 phase cells (Figure 4A-B). Figure 4E better demonstrates protein degradation efficiency as a function of both the cell size and cell cycle stage. In particular, large cells at G1/S transition show the highest protein degradation efficiency (i.e., degradation rates per unit mass). These data suggest that protein degradation rates may be regulated by both cell size and the cell cycle, and possibly directly regulated by the cell cycle machinery or other cell cycle events. We now comment about this and highlight these important questions in the discussion (line 380-392).

9. In Figure 1B, the authors characterize the average cell size changes after rapamycin or SNS-032 treatment. Because cell size is a distribution over a range, simply averaging all cells together may obscure some informational content. It is probably preferable to plot the distribution of cell sizes (e.g. dot plots or violin plots), comparing control and treatment groups.

We thank the reviewer for the suggestion. The previous Figure 1B-C panels were reproduced from our earlier work (Ginzberg et al., 2018, doi.org/10.7554/*eLife*.26957) to illustrate the assay to a general audience. We agree with the reviewer’s suggestion that presenting the distribution of cell sizes can provide additional clarity. However, considering reviewer #3’s recommendation to omit these figures and cite the original publication instead, we have moved these panels to the supplement (now Figure 1—figure supplement 1). For consistency in the figure presentation, we have retained the original figure format as in the published version. We will consult with the editorial team regarding the use and reformatting of these reproduced figures.

10. Figure 1D shows that the nuclear area drops around G1/S when Cdk2 inhibitor is used. However, when they replotted the data by aligning cells based on the timeline relative to G1/S, this phenotype is not obvious anymore (Figure 1E). Please comment on why this is.

We thank the reviewer for this insightful observation.

Figure 1D-E are derived from live-cell imaging experiments and are intended to illustrate the phenomenon of size dependent compensatory growth control. Under CDK2 inhibition, cells spend a longer time in G1 but compensate by reducing their growth rate, ultimately entering the S phase at a size comparable to that of control cells. This conclusion is based on growth rate measurements averaged across the cell cycle.

As the reviewer noted, Figure 1D shows a decrease in the nuclear area as cells approach the G1/S transition under CDK2 inhibition. However, when the data is synchronized based on the G1/S transition (Figure 1E), this decrease seems less apparent. This may be partly due to the two figures being displayed at different timescales.

Nevertheless, while the data allows us to conclude the overall slowdown of the growth rate in CDK2-inhibited cells averaged across the entire cell cycle, the current datasets do not provide sufficient statistical power to conclude whether CDK2-inhibited cells exhibit particularly prominent growth slowdown during a short period around G1/S transition compared to other cell cycle stages. Thus we have decided not to comment further on this aspect in the manuscript. However, this is an intriguing area for future investigation.

11. Page 3, line 112: should the reference be to Figure 1B?

Yes. Thank you. We have now corrected this typo.

Reviewer #3 (Recommendations for the authors):1. While the main conclusion of the paper that larger cells size compensate by increased proteasome-dependent protein turnover is supported by key experiments (Figure 3 D,E; Figure 4 A,E) the link to cell size is largely correlative, as supported by exploiting stochastic size variation in unperturbed cells and treatment with the CDK2 inhibitor in a perturbed context. To my mind, an important control would be to look in the situation where cells become large and size compensation does not occur, i.e., CDK6 inhibitor treated cells. The authors' model predicts that there should be no increase in proteasomal activity in the CDK6 inhibitor-treated cells. It seems likely that authors would have considered this experiment given their published observation that CDK6 inhibition is accompanied by unrestrained growth without size compensation.

We appreciate the reviewer's suggestion and have incorporated measurements with a CDK4/6 inhibitor (Figure 3—figure supplement 4A-C). CDK4/6 inhibition led to a significant increase in cell size. And aligning with the reviewer expectation, while protein degradation rates did increase with CDK4/6 inhibition, this increase was proportional to the increase in cell size. This is in contrast to the phenotype of CDK2 inhibition, where the increase in degradation rates was superlinear and exceeded the expectation based on cell size alone. Taken together, the new data aligns with the reviewer’s suggestion that compensatory degradation is not activated in CDK4/6-inhibited cells that reprograms a larger target size without inducing compensatory growth slowdown. These new results are now incorporated in the revised manuscript line 224-234.

2. One mechanism that could contribute to the increased rate of degradation is abortive translation that requires proteasome mediated clearance of partial translation products. It has long been known that turnover of nascent translated proteins accounts for a substantial fraction of proteasomal flux (~50% if I recall correctly). If translation rate is higher in large cells, then basal protein degradation may also be correspondingly higher. Although the size normalization of K48 chain turnover suggests that additional degradation indeed occurs in large cells, a possible further control to test this idea would be to treat cells with an acute sub-lethal dose of cycloheximide to slow translation followed by measurement of proteasome activity. Alternatively, treatment of cells with a low dose proteasome inhibitor would be predicted to reduce size compensation and allow cells to become larger.

We thank the reviewer for the suggestion. Please refer to our detailed response above to essential point #3.